# Artificial Intelligence—A Tool for Risk Assessment of Delayed-Graft Function in Kidney Transplant

**DOI:** 10.3390/jcm10225244

**Published:** 2021-11-11

**Authors:** Andrzej Konieczny, Jakub Stojanowski, Klaudia Rydzyńska, Mariusz Kusztal, Magdalena Krajewska

**Affiliations:** Department of Nephrology and Transplantation Medicine, Wroclaw Medical University, 50-556 Wroclaw, Poland; jakub.stojanowski@student.umw.edu.pl (J.S.); klaudia.rydzynska@student.umw.edu.pl (K.R.); mariusz.kusztal@umw.edu.pl (M.K.); magdalena.krajewska@umw.edu.pl (M.K.)

**Keywords:** artificial intelligence, machine learning, delayed-graft function, deceased donors, kidney transplantation

## Abstract

Delayed-graft function (DGF) might be responsible for shorter graft survival. Therefore, a clinical tool predicting its occurrence is vital for the risk assessment of transplant outcomes. In a single-center study, we conducted data mining and machine learning experiments, resulting in DGF predictive models based on random forest classifiers (RF) and an artificial neural network called multi-layer perceptron (MLP). All designed models had four common input parameters, determining the best accuracy and discriminant ability: donor’s eGFR, recipient’s BMI, donor’s BMI, and recipient–donor weight difference. RF and MLP designs, using these parameters, achieved an accuracy of 84.38% and an area under curve (AUC) 0.84. The model additionally implementing a donor’s age, gender, and Kidney Donor Profile Index (KDPI) accomplished an accuracy of 93.75% and an AUC of 0.91. The other configuration with the estimated post-transplant survival (EPTS) and the kidney donor risk profile (KDRI) achieved an accuracy of 93.75% and an AUC of 0.92. Using machine learning, we were able to assess the risk of DGF in recipients after kidney transplant from a deceased donor. Our solution is scalable and can be improved during subsequent transplants. Based on the new data, the models can achieve better outcomes.

## 1. Introduction

Delayed-graft function (DGF) after a kidney transplantation (KTx) refers to an acute kidney injury (AKI) requiring at least one dialysis session within the first week after surgery. It is associated with prolonged hospitalization, higher rates of acute rejection, and, therefore, shorter graft survival [1,2,3]. The incidence of DGF is rising due to an increasing employment of kidneys procured from extended criteria donors, caused by organ shortages [4,5]. Based on US data, its prevalence is about 30.8% among deceased donors and it is significantly higher in procurement after circulatory death (DCD), at around 45–55.1% [3].

The ability to predict DGF is crucial in decision-making processes at the time of transplantation, including declining the offer, selecting a recipient with a lower DGF risk, implementing efforts to shorten the cold ischemia time (CIT), or postponing an initiation of treatment with calcineurin inhibitors (CNIs). A clinical tool allowing for the anticipation of DGF might be vital for the outcome. The most widely used tool, with about 70% accuracy, is the Irish risk calculator, which is based on 20 recipient- and donor-derived risk factors; it can identify the most significant aspects such as CIT, a donor’s terminal creatinine concentration, a donor’s body mass index (BMI), procurement after DCD, and a donor’s age [6].

In 1959, Arthur Samuel defined the term “machine learning” as a field of study that gives computers the ability to learn without being explicitly programmed [7]. The premise of machine learning (ML) is to mimic human brain activity, including decision-making processes, such as recognizing or classifying objects. This might be accomplished by optimization and generalization. Optimization means the search for model parameters, and a loss function is as small as these parameters can be. The aim is to keep this value as low as permissible on new data. The model is designed to formulate conclusions from the training data, so that it can generate effective results from the test data. The program gains experience by learning new information and, at the same time, improves its parameters to adapt to the new knowledge. ML algorithms also need relevant data to learn from.

In a classical statistical analysis, independent variables are examined. ML may be more effective when introduced variables are calculated from other variables, such as BMI, which is derived from height and weight, as well as partially dependent parameters. Results received from other parameters may facilitate the finding of an optimal model with a sufficiently discriminant power. However, their use increases the complexity and computational cost of the algorithm [8].

In our work, we verified the factors that may be responsible for the occurrence of DGF with the use of information technology, particularly the branch of artificial intelligence called machine learning.

## 2. Materials and Methods

### 2.1. Data Collection

The study was conducted in a single transplant center. Retrospective patient data, both donors and recipients, who underwent KTx between 2012 and 2018, were included in the study. DGF was defined as the need for at least one dialysis during the first week after transplantation.

The following donor-derived data were collected: age, gender, weight, height, serum creatinine concentration (sCr) directly before procurement, minimal sCr during hospitalization in an intensive care unit (ICU), presence of diabetes (DM) or arterial hypertension (HTN) in medical history, cause of death, the length of stay in ICU, and the need for catecholamine use. We considered donors’ minimal sCr (sCr min) and sCr before procurement as indicators of organ damage [9]. The estimated glomerular filtration rate (eGFR) was determined by the modification of diet in renal disease (MDRD) formula [10]:$${\mathrm{eGFR}}_{\mathrm{MDRD}}=175\cdot {\mathrm{sCr}}^{-1.154}\cdot {\mathrm{Age}}^{-0.203}\cdot 0.742\;\mathrm{if}\;\mathrm{female}$$

Accordingly, minimal eGFR and eGFR before procurement, which were sCr derived according to the above formula, served as alternative indicators of damage to the transplanted organ.

Based on donor-derived data, the kidney donor risk index (KDRI) and the kidney donor profile index (KDPI), at the time of procurement, were assessed using the organ procurement and transplantation network (OPTN) online calculator.

KDRI is an indicator that measures the risk of failure after a KTx. The parameter combines 10 donor-derived variables: age, height, weight, ethnicity/race, presence of HTN, DM, cause of death, sCr, HCV status, and donation after circulatory death [11,12]. KDPI stands for cumulative risk of kidney donor failure and allows one to effectively assess the risk of DGF; it is a good indicator of transplant and recipient survival, and it also discriminates against cardiac causes of death with cerebrovascular death [12,13].

Recipient data were also included: age, gender, height, weight, BMI, residual diuresis, presence of DM or HTN, type of renal replacement therapy (RTT) before transplantation and its duration, the use of basiliximab as an induction therapy, the KTx number, length of stay (LOS), and sCr at discharge. The estimated post-transplant survival (EPTS) for each recipient was calculated, using 4 variable parameters: age, duration of RTT, presence of DM, and history of previous solid organ transplantations (kidney, pancreas, liver, lung, heart, intestine) [14].

### 2.2. Statistical Scoring

Our task was to find a model that, using the available database, had the best performance. Models were evaluated by 10-fold cross validation against accuracy. We based the final assessment of effectiveness on AUROC, accuracy, precision, recall, and F1-Score. Derivations are counted for each of the classes (0 or 1) and further expressed as arithmetic mean and weighted mean.

Accuracy:$$\frac{\mathrm{TP}+\mathrm{TN}}{\mathrm{TP}+\mathrm{TN}+\mathrm{FP}+\mathrm{FN}}$$

Precision or positive predictive value (PPV):$$\frac{\mathrm{TP}}{\mathrm{TN}+\mathrm{FP}}$$

Recall or sensitivity:$$\frac{\mathrm{TP}}{\mathrm{TP}+\mathrm{FN}}$$

F1-Score:$$\frac{2\mathrm{TP}}{2\mathrm{TP}+\mathrm{FP}+\mathrm{FN}}$$

Abbreviations: TP—true positives, FP—false positives, TN—true negatives, and FN—false negatives.

### 2.3. Machine Learning Models

In our previous work, we analyzed 5 types of classifiers. This time, we focused on the two most effective ones: ANN and random forest classifiers [15].

The random forest (RF) classifier creates decision trees for the training set samples and finally averages the results. This helps to avoid overfitting, i.e., too rigid an adjustment of the added training sessions and a drop in efficiency in the test data. A practical feature of this method enables the visualization of data in the form of a decision tree, which might sometimes be very complex. It may be clearly seen how the decision-making process was carried out by the algorithm.

An artificial neural network is a collection of units, called neurons, that are connected to each other in such a way as to form input, hidden layers, and output layers. The way in which neurons communicate mimics the connection of neurons in the brain. Activation function for the hidden layer is ReLu, the rectified linear unit function, which returns 0 for negative values and input value for non-negative: f (x) = max (0, x). Neural networks are more complex and have more coefficients inscribed in the relevant elements of their structure. Each neuron in a single MLP layer stores the values of the connection weights with the perceptrons of the previous layer, i.e., it has connections with each neuron of the previous layer. The advantage of neural networks is that they may be learned from new data without starting from scratch. By partial fitting of new data, the existing neural network can be overwritten with new weights. At the same time, complexity and sensitivity to data normalization are serious drawbacks of neural networks. Random forest can be built faster than scratch on new data, and it is not sensitive to data scaling and normalization.

The complexity of algorithms significantly affects their practical application, especially in the process of building predictive models. The notation with a big O is used, which means that the complexity of the model is not greater than a certain mathematical function multiplied by a positive real number. Complexity may refer to the time required to build the model, the computer memory consumed, or the amount of time a program runs until a result is obtained. The complexity of training the random forest classifier is O (M·m·n·log(n)), where M is the number of decision trees in the random forest, m is the number of variables, and n is the number of samples in the training set [16]. This means that reducing the number of input parameters will shorten training time by half. Doubling the number of samples in the 1000 patient database will cause the algorithm to train for approximately 2.2 times longer. An MLP neural network has complexity O (n·m·o·i·h1·h2), where n is the number of samples in the training set, m is the number of input features, and “o” is the number of predicted classes, e.g., absence or presence of DGF. The sizes of the hidden layers are h1 and h2, respectively, and they denote the number of iterations leading to the best model [17,18]. This means that scaling a model from 25 neurons in 2 hidden layers to 125 in each of them increases the training complexity 25-fold.

The initial database was randomly divided into two sets: training and testing, in a ratio of 80:20. At each step of the algorithm, the program constructed a subset regarding the analyzed variables in a recursive manner. The original number of variables was recursively reduced towards the optimal subset. In each algorithm loop, the program built a model based on training data and checked its effectiveness. The training set was used to find the best model hyperparameters using 10-fold cross validation against accuracy. The model was selected based on cross validation, and statistics against test data, including AUROC, accuracy, precision, recall, and F1-Score, were saved to program memory. After checking all subsets, the program sorted the results and selected the best models. Figure 1 illustrates the block diagram of the recursive algorithm allowing the development of the models.

## 3. Results

### Study Population Baseline Characteristics

In our study, we included data of 157 organ recipients aged 19–72 years (50.55 ± 13.08) and 88 donors aged 18–69 years (46.38 ± 14.02). From each of the 69 donors, two kidneys were procured, and one kidney was procured from 19 donors.

All evaluated parameters and variables are listed in Table 1.

The input database was randomly divided into two cohorts: training and testing. Most donors (69 out of 88) provided a total of 138 records for each transplant procedure and 19 donors provided individual records. In total, the database contains 157 records with data. The characteristics of the divided groups are described in Table 2.

All possible subsets of the input parameters were recursively selected, with a minimum size of two parameters. The best performance was achieved by models based on at least four key parameters: donor’s BMI, recipient’s BMI, recipient–donor weight difference, and donor’s eGFR before procurement using random forest classifier and MLP with an accuracy of 84.38%. Models with fewer input variables were completely ineffective. The best models we found required the above mentioned four variables plus EPTS, KDRI, KDPI, recipient’s gender, or recipient’s age, and the result was the random forest and MLP models, which are summarized in Figure 2. 

Characteristically, the top models have a similar set of input features, and differences in performance are similar. The best random forest classifier models require the following input features to achieve the discriminant power of a given AUC of 0.91 (AUC 0.92): donor’s BM, recipient’s BMI, recipient–donor weight difference, and donor’s eGFR, as well as additional variables: a pair of EPTS% and KDRI or a triplet of KDPI recipient’s gender, recipient’s age.

The first model is a classifier based on the random forest method. Input parameters included: donor’s BMI, recipient’s age, recipient’s gender, donor’s eGFR before procurement, KDPI, recipient–donor weight difference, and recipient’s BMI, with an accuracy of 93.75%, precision of 0.9375 and recall of 0.9375. Figure 3 illustrates the decision tree of this model. 

The nodes contain conditions, the fulfillment of which means moving to the left child branch in the decision tree. Otherwise, the right child node is selected. The intensity of the color means that the knot is class-uniform. End nodes uniquely defining y one of the end labels, i.e., 0 or 1, are completely homogeneous. Each node is a data break point. The functional composition of such divisions is the fundamental of the classifier’s operation on data. For example, in a first step, the condition is checked: if KDPI is less or equal 15.50, then the model judges that no DGF will occur; otherwise, the cascade of conditions leading to the corresponding end states is checked. This model achieved an AUC of 0.91, showed in Figure 4.

Random forest classifier with input features: donor’s BMI, donor’s eGFR before procurement, EPTS, KDRI, recipient–donor weight difference, and recipient’s BMI. The second model with the highest scores is shown in Figure 5.

This model achieved an accuracy of 93.75%, precision of 0.9375, and recall of 0.9375 and achieved slightly better discriminant power than the previous one (presented in Figure 6).

Random forest classifier with input features: donor’s BMI, donor’s eGFR before procurement, EPTS, recipient–donor weight difference, recipient’s BMI, with an accuracy of 87.50%, precision of 0.8718 and recall of 0.8750, as shown in Figure 7.

This classifier achieved a slightly worse discriminating power than the previous ones, the performance is summarized in Figure 8.

Random forest classifier with input features: donor’s BMI, donor’s eGFR before procurement, recipient–donor weight difference, recipient’s BMI, with an accuracy of 84.38%, precision of 0.8514 and recall of 0.8438. The classifier is illustrated by the decision graph in Figure 9.

The performance of the model is summarized in Figure 10.

MLP with 6 neurons in first hidden layer and 37 neurons in the second, with input features: donor’s BMI, donor’s eGFR before procurement, recipient–donor weight difference, recipient’s BMI, with an accuracy of 84.38%, precision of 0.8734 and recall of 0.8438. The performance of the artificial neural network is summarized in Figure 11.

The matrix in Figure 12 shows the accuracy of ANN with input features: donor’s BMI, donor’s eGFR before procurement, recipient–donor weight difference, recipient’s BMI, with an accuracy of 84.38%. 

For a randomly selected testing subset, depending on the selection of hyperparameters, the number of neurons in the first layer is on the vertical axis and the number of neurons in the second layer is on the horizontal axis. Higher accuracy is marked in green, while worse is yellow and red. The best results are for an ANN with fewer neurons in the first layer and more in the second, and vice versa. On the other hand, equal-dimensional layers have a worse performance in this representation. However, this is not the subject of this analysis but instead an observation worthy of further interest.

## 4. Discussion

Different logistical solutions in the allocation of permanent organs, variability of donors (e.g., accepting expanded criteria), and recipient conditions in a given geographical region are sufficient reasons to assess the risk of DGF in a single transplant center. To assess the risk of DGF, we employed data mining and machine learning experiments. Random forest classifiers and artificial neural networks called multi-layer perceptron showed an accuracy of 84.38% and an AUC 0.84. The model additionally implementing the donor’s age, gender, and the kidney donor profile index accomplished an accuracy of 93.75% and an AUC of 0.91. The other configuration with the estimated post-transplant survival (EPTS) and the kidney donor risk profile (KDRI) achieved an accuracy of 93.75% and an AUC of 0.92.

The growing shortage of organs for transplantation requires procurement from expanded criteria donors (ECD). ECD might be described for any donor over the age of 60, or donors over the age of 50, meeting two out of three criteria: history of hypertension, final sCr > 1.5 mg/dL, or death due to cerebrovascular disease [5,19]. Kidney transplantation from ECD carries additional risk factors for graft failure, including DGF [5]. Risk modeling using artificial intelligence opens an area for the use of organs from ECD in the face of increased organ demand and allows use to manage the risk of failure. Non-linear identification of risk factors and their multidimensional analysis may improve the survival of grafts in the future and reduce the risk of graft failure after transplantation from a deceased donor.

Machine learning predictive models allow for the effective differentiation of end states and practical application of discriminant power in the clinic. In our work, we checked two machine learning techniques through by random forest and artificial neural network models. Similar models were described by other authors [20,21,22,23]. Neural networks aimed at assessing the occurrence of DGF achieved various performances, but clearly indicated the practicality and clinical usefulness of modern computer techniques. Donor BMI and recipient BMI are among the most important predictors of the neural network model with AUROC 0.886 [20]. Additionally, clinical trials based on classical analysis prove their significance [2,4,13]. Success in creating computer models requires two keys: the first is good quality data and the second is an optimized model suitable for the data [24]. Modern computers make it possible to check many models, with different parameters and structures in a short time. There are many machine learning techniques, and models that are theoretically simpler and less complex can perform significantly better than advanced ones. Artificial neural networks can achieve good performance with AUROC 0.732 or AUROC 0.7595 [21,23], but a simpler model based on linear SVM achieves an AUROC of 0.843 [22]. The question is whether it is worth investing in the complexity of the model. Our previous study showed that this depends on the choice of input data, both the data type and the combination of the input parameters [15]. Naturally, neural networks are more complex and can offer better performance, but simpler solutions can be just as effective. The choice of method can be left to individual experience because a modern personal computer is able to check thousands of models quickly, with neural networks being checked much slower and requiring more training time.

In our paper, we refer to the commonly accepted variables that are indicated as risk factors of DGF. The size of the original database, considering prognostic factors, was so large that it provided great opportunities for analyzing data, using machine learning, and selecting clinically significant factors. We checked many models with different settings and input features, and the best models did not implement the CIT variable; although, according to this publication, it is a crucial factor for the risk assessment of DGF. [25,26,27] None of the models using CIT as a variable achieved an accuracy of more than 81.25%. The top model that applied CIT at the input reached an AUROC of 0.78, an accuracy of 81.25%, precision of 0.8318 (0.91 for non-DGF prediction and 0.60 for DGF presence), and recall of 0.8125 (0.83 and 0.75, respectively). We explain this discrepancy by the difference in methodology between classical odds ratios and random forest algorithms. Nevertheless, other key parameters are confirmed both in our work and in others.

Our study was retrospective and limited due to the size of the population. Models were prepared in a single-center study. The advantage of this is that the methods used allow for the scaling of machine learning techniques to another population, one that might be diverse in terms of ethnicity. Neural networks are so complex that their use outside of a computer program is a futile endeavor. It is a kind of black box into which we put a patient’s characteristics and draw a ready conclusion. Random forest classifiers can be illustrated graphically and, despite their complex form, can be used with paper and pencil. Of course, this is only to illustrate the clarity of this technique compared to artificial neural networks.

## 5. Conclusions

Using machine learning, we were able to assess the risk of DGF in recipients after kidney transplant from a deceased donor. Our solution is scalable and can be improved during subsequent transplants. Based on the new data, the models can achieve better outcomes in terms of what is feasible in even single transplant center.

## Figures and Tables

**Figure 1 jcm-10-05244-f001:**
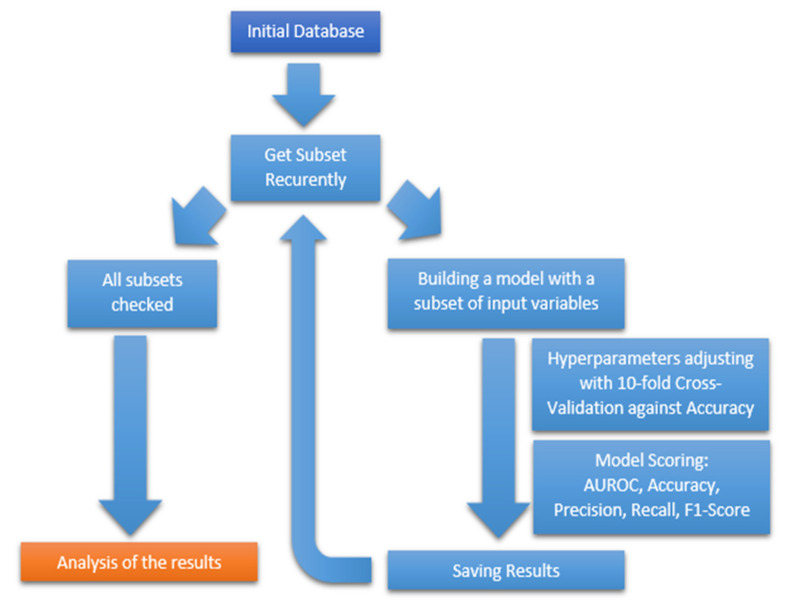
The database is divided into a training and testing set. The training set is the initial database from which we recursively removed the columns, thus changing the size of the input data vector (the number of input features). The model was then built and hyperparameters were adjusted. The performance was saved to a file for later analysis. The analyzed subsets have at least 2 input variables.

**Figure 2 jcm-10-05244-f002:**
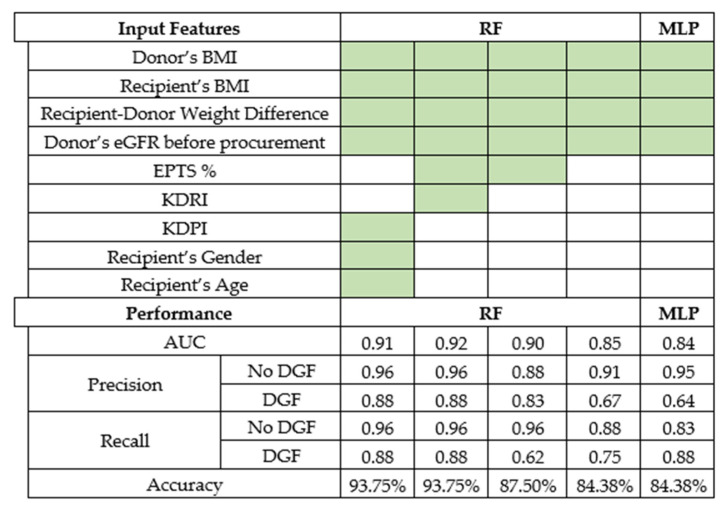
Various sets of input features and their performance statistics: all top models needed a database of 4 variables: donor’s BMI, recipient’s BMI, recipient–donor weight difference, and donor’s eGFR before procurement. A colored cell means that the variable from the input features column was used by the RF or MLP model. Precision and recall were measured separately for each class: DGF occurrences or no DGF. Some models needed additional variables, as shown in the table. More or fewer variables influenced the performance.

**Figure 3 jcm-10-05244-f003:**
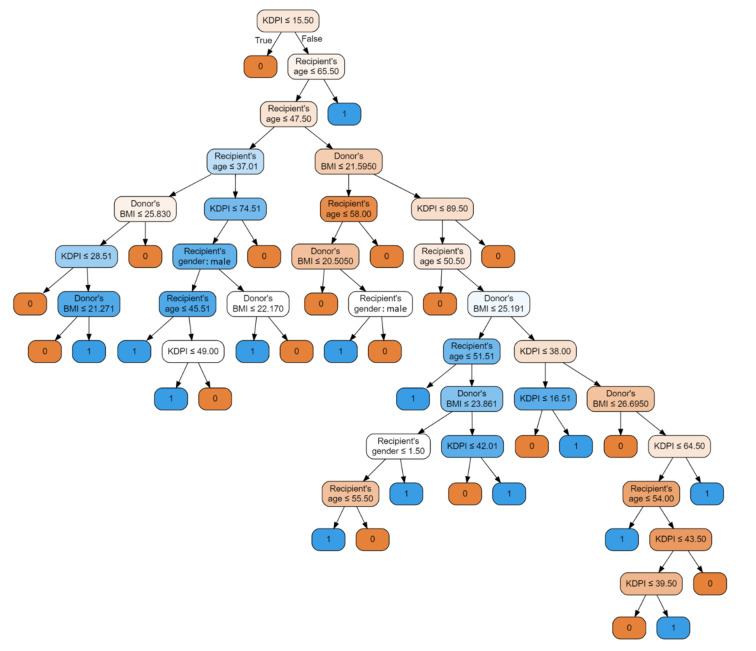
Random forest classifier illustrated with a decision tree graph. Each node has a condition; if the condition is met, it goes to the child branch on the left, otherwise to the right branch. The more uniform the color, the clearer the node is in relation to the samples it contains. Input features include donor’s BMI, recipient’s age, recipient’s gender, donor’s eGFR before procurement, KDPI, recipient–donor weight difference, recipient’s BMI.

**Figure 4 jcm-10-05244-f004:**
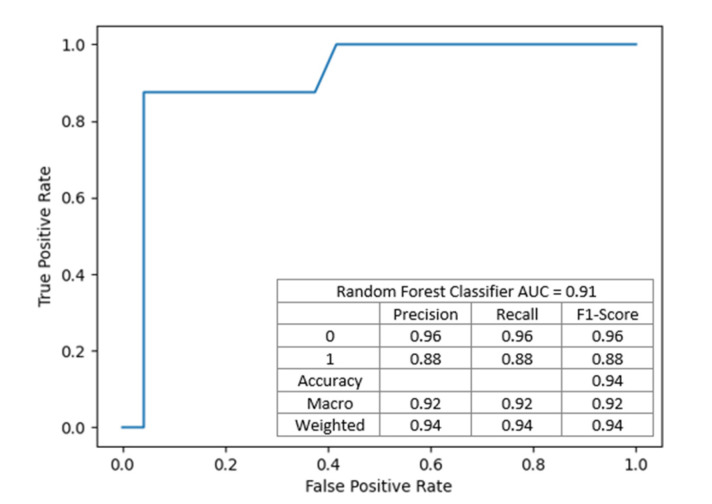
The model with the best performance has 7 input variables allowing efficiently discriminate (AUC = 0.91) the occurrence and non-occurrence of DGF in a patient after transplantation.

**Figure 5 jcm-10-05244-f005:**
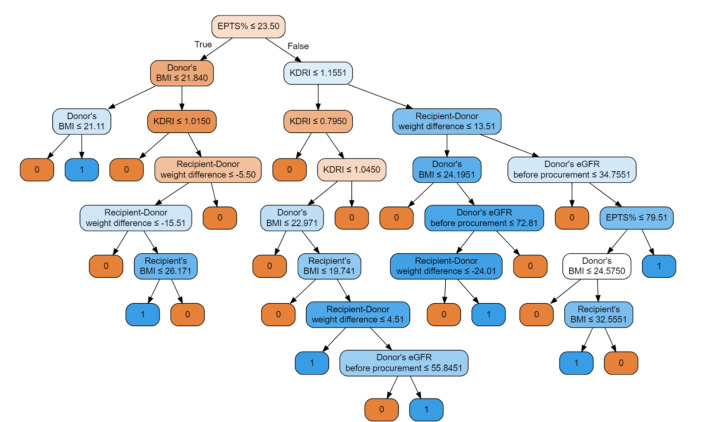
Random forest classifier with input features including six parameters: donor’s BMI, donor’s eGFR before procurement, EPTS, KDRI, recipient–donor weight difference, recipient’s BMI. The model has 4 input parameters that are also significant for the previous model.

**Figure 6 jcm-10-05244-f006:**
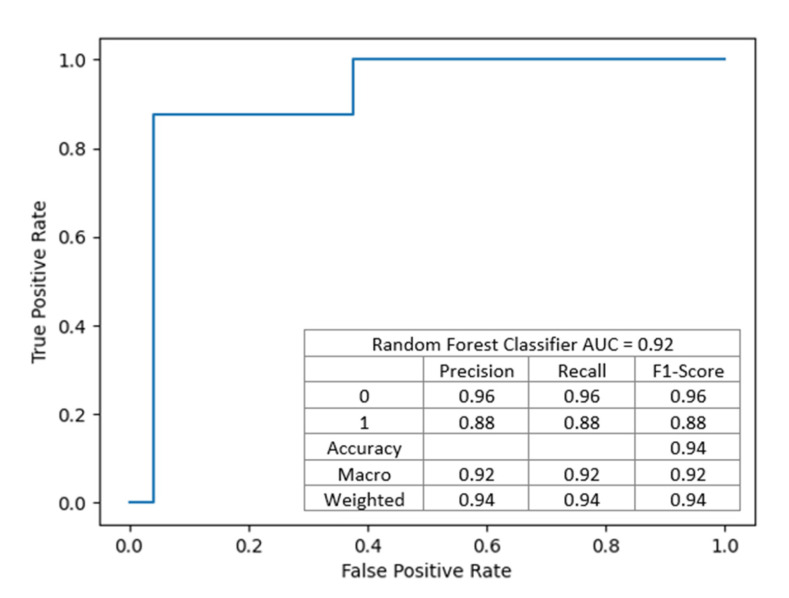
This model achieved a slightly better differentiation power of classes given by AUC = 0.92.

**Figure 7 jcm-10-05244-f007:**
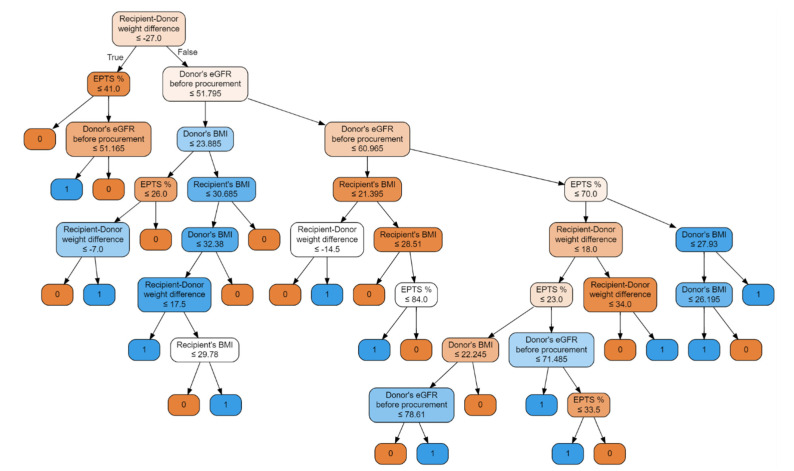
Random forest classifier with five input features: donor’s BMI, donor’s eGFR before procurement, EPTS, recipient–donor weight difference, recipient’s BMI.

**Figure 8 jcm-10-05244-f008:**
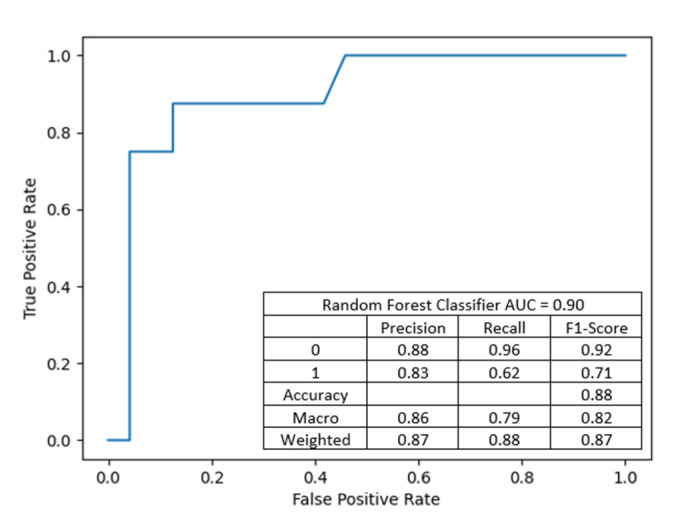
The model classifies patients slightly worse in terms of prediction of DGF occurrence. Despite good general parameters, it has a low sensitivity (0.62) in relation to DGF occurrence.

**Figure 9 jcm-10-05244-f009:**
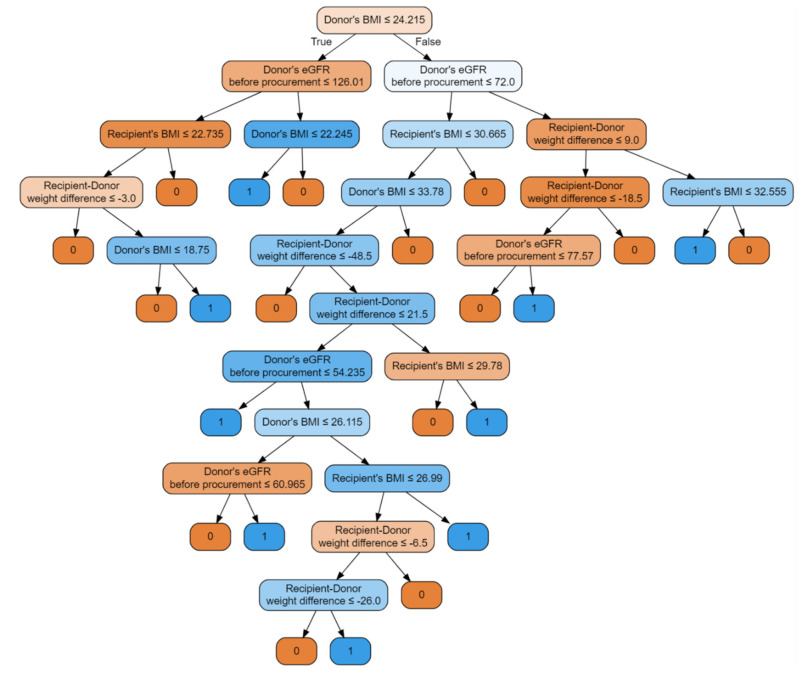
Random forest classifier with input features: donor’s BMI, donor’s eGFR before procurement, recipient–donor weight difference, recipient’s BMI.

**Figure 10 jcm-10-05244-f010:**
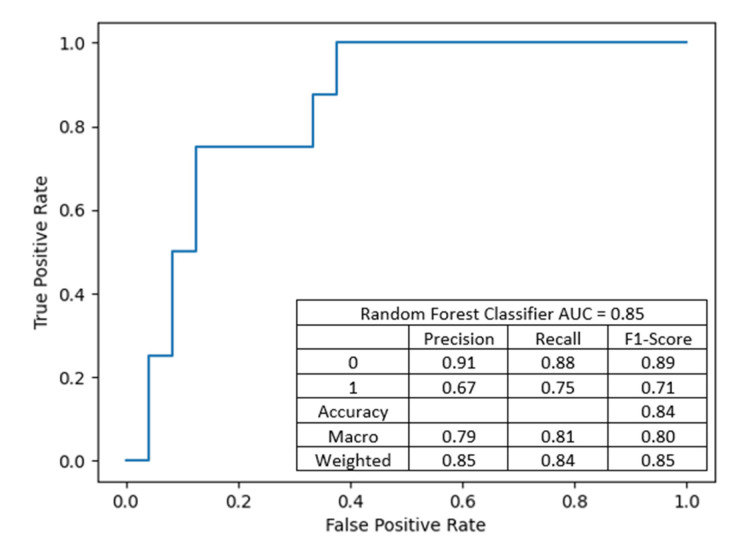
This classifier has a lower discriminant power but better DGF prediction sensitivity than the previous model.

**Figure 11 jcm-10-05244-f011:**
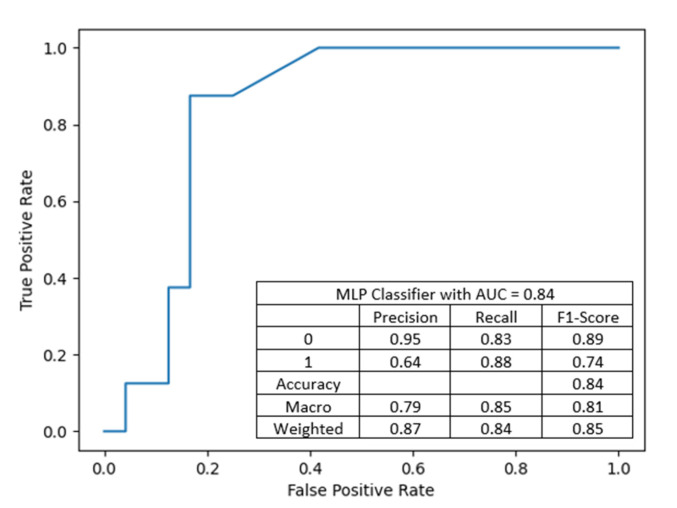
An artificial neural network based on multi-layer perceptron: the classifier is the only one with greater sensitivity to the presence of DGF than to the absence of DGF. The combination of this model and the previous one can still be a prognostic tool.

**Figure 12 jcm-10-05244-f012:**
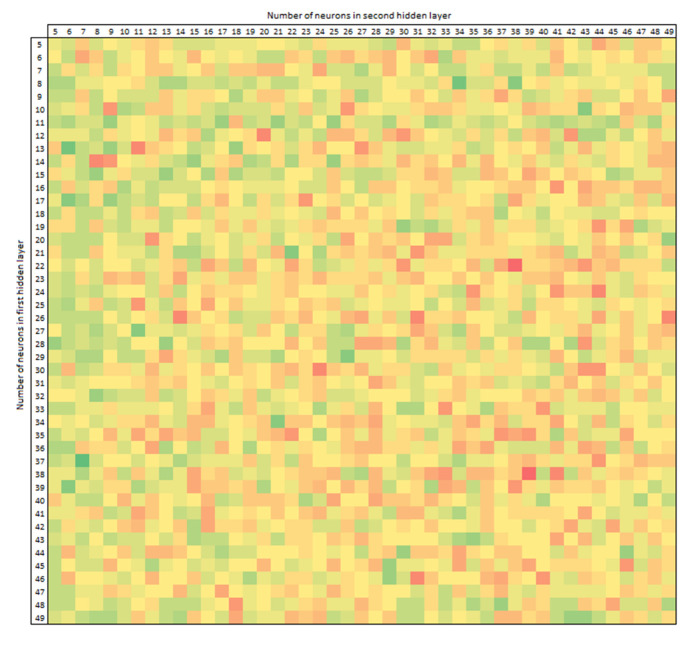
The number in the vertical row is the number of neurons in the first hidden layer, and the number in the horizontal row is the number in the second hidden layer of neurons in an artificial neural network made of perceptrons. The greener the color, the greater the accuracy of the model; the redder, the worse the accuracy.

**Table 1 jcm-10-05244-t001:** Basic characteristics of the population.

Parameter	Population (Mean ± Standard Derivation (SD) and Range from Minimal to Maximal Value) Categorical (If Applicable) Not Included in Program Analysis
donor’s age (years)	46.38 ± 14.02 (18 ÷ 69)
donor’s gender (male/female)	52/36 (59.1%/40.9%)
donor’s weight (kg)	77.34 ± 15.8 (41 ÷ 145)
donor’s height (cm)	172.82 ± 9.71 (152 ÷ 200)
donor’s BMI (kg/m^2^)	25.81 ± 4.46 (16.02 ÷ 46.81)
donor’s sCr before procurement (mg/dL)	1.24 ± 0.61 (0.36 ÷ 3.63)
donor’s eGFR before procurement (mL/min/1.73 m^2^)	78.03 ± 42.91 (18.59 ÷ 214.76)
donor’s sCr min (mg/dL)	1.06 ± 0.48 (0.36 ÷ 3)
donor’s eGFR min (mL/min/1.73 m^2^)	89.0 ± 44.23 (23.51 ÷ 214.76)
donor’s DM (No/Yes)	83/5 (94.32%/5.68%)
donor’s HTN (No/Yes)	65/23 (73.86%/26.14%)
cause of donor’s death (head trauma/cerebrovascular/anoxia)	32/38/18 (36.36%/43.18%/20.46%)
KDPI	52.8 ± 27.89 (2 ÷ 99)
KDRI	1.11 ± 0.37 (0.59 ÷ 2.24)
catecholamines use (No/Yes)	9/79 (10.23%/89.77%)
catecholamines number (0/1/2/3)	9/62/16/1 (10.23%/70.45%/18.18%/1.14%)
length of stay in the ICU (days)	5.43 ± 3.61 (1 ÷ 22)
recipient’s age (years)	50.55 ± 13.08 (19 ÷ 72)
recipient’s gender (Male/Female)	112/45 (71.3%/28.7%)
recipient weight (kg)	75.49 ± 13.45 (47 ÷ 105)
recipient’s height (cm)	172.01 ± 9.11 (145 ÷ 196)
recipient’s BMI (kg/m^2^)	25.43 ± 3.61 (18.31 ÷ 33.36)
recipient’s residual diuresis (mL/24 h)	778.34 ± 645.13 (0 ÷ 3000)
recipient’s HTN (No/Yes)	3/154 (1.9%/98.1%)
recipient’s DM (No/Yes)	122/35 (77.7%/22.3%)
type of RRT (hemodialysis/peritoneal dialysis)	140/17 (89.2%/10.8%)
RTT duration (years)	2.27 ± 1.67 (0 ÷ 7)
KTx number (1st/2nd)	141/16 (89.8%/10.2%)
EPTS (%)	35.92 ± 27.52 (1 ÷ 97)
number of HLA mismatches (0/1/2/3/4/5/6)	4/10/32/49/41/19/2 (2.5%/6.4%/20.4%/31.2%/26.1%/12.1%/1.3%)
CIT (h)	20.29 ± 6.63 (1 ÷ 36)
immunosuppression (cyclosporin/tacrolimus)	17/140 (10.8%/89.2%)
basiliximab in induction therapy (No/Yes)	131/26 (83.4%/16.6%)
DGF duration (days)	3.55 ± 5.48 (0 ÷ 22)
LOS (days)	22.14 ± 10.62 (10 ÷ 69)
sCr at discharge (mg/dL)	1.51 ± 0.43 (0.69 ÷ 2.56)
eGFR at discharge (mL/min/1.73 m^2^)	53.46 ± 17.33 (24.89 ÷ 102.73)
DGF (No/Yes)	97/60 (61.8%/38.2%)

BMI—body mass index; sCr—serum creatinine concentration; eGFR—estimated glomerular filtration rate; DM—diabetes, HTN—arterial hypertension; KDRI—kidney donor risk index; KDPI—kidney donor profile index; ICU—intensive care unit; KTx—kidney transplantation, DGF—delayed-graft function, LOS—length of stay; HLA—human leukocyte antigens; CIT—cold ischemia time; EPTS—estimated post-transplant survival.

**Table 2 jcm-10-05244-t002:** Baseline characteristics of the donors and recipients enrolled in the cohorts. Most donors (69 out of 88) provided a total of 138 records for each transplant procedure and 19 donors provided individual records. Training and testing sets are described by mean ± standard derivation (SD) and range from minimal to maximal value or as categorical, if applicable.

Patients’ Parameters (N)	Study Cohort (Training Set) *n* = 125	Test Cohort (Testing Set) *n* = 32
donor’s age (years)	46.35 ± 13.68	45.56 ± 15.18
donor’s gender (male/female)	73/52	21/11
donor’s weight (kg)	77.28 ± 16.41	79.34 ± 14.5
donor’s height (cm)	172.36 ± 9.9	175.31 ± 9.32
donor’s BMI (kg/m^2^)	25.92 ± 4.66	25.75 ± 3.94
donor’s sCr before procurement (mg/dL)	1.25 ± 0.63	1.24 ± 0.54
donor’s eGFR before procurement (mL/min/1.73 m^2^)	78.22 ± 43.32	77.49 ± 40.62
donor’s sCr min (mg/dL)	1.05 ± 0.47	1.08 ± 0.49
donor’s eGFR min (mL/min/1.73 m^2^)	90.64 ± 45.74	87.01 ± 38.58
donor’s DM (No/Yes)	119/6	31/1
donor’s HTN (No/Yes)	92/33	25/7
KDPI	52.83 ± 27.21	50.38 ± 28.76
KDRI	1.1 ± 0.36	1.07 ± 0.35
cause of donor’s death (head trauma/cerebrovascular/anoxia)	44/52/29	14/13/5
catecholamines use (No/Yes)	13/112	4/28
catecholamines number (0/1/2/3)	13/87/23/2	4/22/6/0
length of stay in the ICU (days)	5.35 ± 3.51	5.81 ± 4.37
EPTS (%)	35.98 ± 27.03	35.66 ± 29.81
recipient’s age (years)	51.26 ± 12.74	47.81 ± 14.24
recipient’s gender (male/female)	89/36	23/9
number of HLA mismatches	3.14	3.13
CIT (h)	20.34 ± 6.83	20.06 ± 5.87
immunosuppression (cyclosporine/tacrolimus)	14/111	3/29
basiliximab in induction therapy (No/Yes)	103/22	28/4
recipient’s height (cm)	171.46 ± 8.92	174.16 ± 9.66
recipient weight (kg)	74.55 ± 13.03	79.16 ± 14.65
recipient’s BMI (kg/m^2^)	25.28 ± 3.55	26.01 ± 3.83
recipient’s residual diuresis (mL/24 h)	772.8 ± 634.67	800 ± 694.68
recipient’s HTN (No/Yes)	2/123	1/31
recipient’s DM (No/Yes)	100/25	22/10
type of RRT (hemodialysis/peritoneal dialysis)	109/16	31/1
RTT duration (years)	2.27	2.34
KTx number (1st/2nd)	113/12	28/4
DGF (No/Yes)	79/46	18/14

## Data Availability

Data are contained within the article.

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
