# Peer review of "Artificial Intelligence—A Tool for Risk Assessment of Delayed-Graft Function in Kidney Transplant"

_jcm, 2021, doi:10.3390/jcm10225244_

Round 1
Reviewer 1 Report
The autors of the paper entitled Artificial intelligence - a tool for risk assessment of delayed-2 graft function in kidney transplant, address the problem of DGF in post renal transplantation using artificial intelligence techniques, through data mining and machine learning experiments, to foster the prediction of the development of DGF in a population of transplant recipients coupled to the data of their donors and their risk factors. Authors show DGF predictive models, based on Random Forest Classifiers (RF) and artificial neural net- works called Multi-Layer Perceptron (MLP). All designed models, had 4 common input parameters, 15 determining the best accuracy and discriminant ability: donor's eGFR, recipient's BMI, donor's BMI, 16 and recipient-donor weight difference. RF and MLP designs, using these parameters, achieved Ac-17 curacy of 84.38% and Area Under Curve (AUC) 0.84. The model additionally implementing the do-18 nor's age, gender, and the Kidney Donor Profile Index (KDPI) accomplished Accuracy of 93.75% 19 and AUC 0.91. The other configuration with the Estimated Post Transplant Survival (EPTS) and the 20 Kidney Donor Risk Profile (KDRI), achieved accuracy of 93.75% and AUC 0.92. Gli autori concludono che Using machine 21 learning we were able to assess the risk of DGF, in recipients after kidney transplant, from a de-22 ceased donor.
The application of these techniques, in an era in which the use of kidney donation with expanded criteria donors is increasingly frequent due to the progressive aging of the population, represent innovative allocation tools, that deserve attention for future extended applications , in these as well as in other fields of medicine. The work gives the novelty of the proposed methodology and gives the particular field of application.
Author Response
We appreciate an effort made by Reviewer and are grateful for all comments.
Reviewer 2 Report
Very interesting and original article, on how to apply the machine learning models in evaluating the risk of DGF after kidney transplantation.
I have only few comments and objections to the Authors:
- the most accurate models include donor's BMI, recipient's age, recipient's gender, donor's eGFR before procurement, KDPI, recipient-donor weight difference, recipient's BMI or donor’s BMI, donor’s eGFR before procurement, EPTS, KDRI, recipient-donor weight difference, recipient’s BMI as input parameters. Though the Authors aim was to evaluate the risk of DGF before KT and most likely during donor evaluation in order to select the most appropriate recipient, their models never include the cold ischemic time (CIT) or at least an expected CIT, according to donor hospital location etc... Indeed CIT is the most important and significant predictor of DGF in kidney transplantation and every additional hour of cold ischemia increases the odds for developing DGF by 8%
Kox J, Moers C, Monbaliu D, et al. The Benefits of Hypothermic Machine Preservation and Short Cold Ischemia Times in Deceased Donor Kidneys. Transplantation. 2018;102(8):1344-1350. doi:10.1097/TP.0000000000002188
Debout A, Foucher Y, Trébern-Launay K, et al. Each additional hour of cold ischemia time significantly increases the risk of graft failure and mortality following renal transplantation. Kidney Int. 2015;87(2). doi:10.1038/ki.2014.304
Therefore, my question to the Authors is the following: is there a model they can develop that can include CIT or at least an expected CIT?
- Table 2 should be improved in clarity
- Please correct the following phrase in line 83 and 84: "using on-line using the Organ Procurement and Transplantation Network (OPTN) online calculator"
Author Response
Very interesting and original article, on how to apply the machine learning models in evaluating the risk of DGF after kidney transplantation.
Thank You for your comments.
I have only few comments and objections to the Authors:
- the most accurate models include donor's BMI, recipient's age, recipient's gender, donor's eGFR before procurement, KDPI, recipient-donor weight difference, recipient's BMI or donor’s BMI, donor’s eGFR before procurement, EPTS, KDRI, recipient-donor weight difference, recipient’s BMI as input parameters. Though the Authors aim was to evaluate the risk of DGF before KT and most likely during donor evaluation in order to select the most appropriate recipient, their models never include the cold ischemic time (CIT) or at least an expected CIT, according to donor hospital location etc... Indeed CIT is the most important and significant predictor of DGF in kidney transplantation and every additional hour of cold ischemia increases the odds for developing DGF by 8%
Kox J, Moers C, Monbaliu D, et al. The Benefits of Hypothermic Machine Preservation and Short Cold Ischemia Times in Deceased Donor Kidneys. Transplantation. 2018;102(8):1344-1350. doi:10.1097/TP.0000000000002188
Debout A, Foucher Y, Trébern-Launay K, et al. Each additional hour of cold ischemia time significantly increases the risk of graft failure and mortality following renal transplantation. Kidney Int. 2015;87(2). doi:10.1038/ki.2014.304
Therefore, my question to the Authors is the following: is there a model they can develop that can include CIT or at least an expected CIT?
The models implementing cold ischemia time (CIT) variable did not achieve accuracy of more than 81.25%. The best model applying CIT reached AUROC 0.78, Accuracy 81.25%, Precision 0.8318 (0.91 for non-DGF prediction and 0.60 for DGF presence), Recall 0.8125 (0.83 and 0.75 respectively). We do not underestimate the crucial role of CIT in DGF pathology but models implementing other factors achieved better accuracy. We explain this discrepancy by the difference in methodology between classical odds ratios and random forest algorithms.
- Table 2 should be improved in clarity
The Table 2 was rearranged.
- Please correct the following phrase in line 83 and 84: "using on-line using the Organ Procurement and Transplantation Network (OPTN) online calculator"
The mistake was corrected.
Reviewer 3 Report
Konieczny et al. evaluated two machine learning algorithms (Random Forest Classifiers and Multi-Layer Perceptron) to predict delayed graft function in kidney transplant patients. I have some comments and suggestions for this study.
- Good prediction for the delay graft function by machine learning algorithms (Random Forest Classifiers and Multi-Layer Perceptron) was found. However, the top important factors were not described or mentioned in this study. We can realize a good prediction from the RF or ANN. However, we also need an explainable machine learning algorithm for the clinical application.
- In the results section, the author mentioned 157 organ recipients. Among the total of 88 donors, 71 donors provided two kidneys, and 17 donors provided one kidney. However, there were 159 kidneys for the recipients (142 + 17 = 159). A discrepancy was noted. Please clarify.
- How many features should be imputed for the models? A continuous variable could be divided into category variables. For example, the BMI could be divided into underweight, normal, overweight, and obesity. Based on the eGFR level, the renal function status could be divided into normal kidney function and different stages of CKD (stage 3-5).
- Several important variables were not included in this study. For example, the time interval from donor to recipient and the time interval from donor’s brain death to the surgery could be considered. For the donor’s parameters, the comorbidity index or risk score for the severity of ICU hospitalization could be considered. For the recipient’s parameters, ABO mismatches, etiology of ESKD, dialysis parameter (kt/v, urea clearance rate, electrolyte, ca/p/PTH, baseline hemoglobin level, albumin, nutrition status), surgical time for kidney transplantation, and time interval from post-surgery to withdrawal from dialysis therapy.
Author Response
Konieczny et al. evaluated two machine learning algorithms (Random Forest Classifiers and Multi-Layer Perceptron) to predict delayed graft function in kidney transplant patients. I have some comments and suggestions for this study.
- Good prediction for the delay graft function by machine learning algorithms (Random Forest Classifiers and Multi-Layer Perceptron) was found. However, the top important factors were not described or mentioned in this study. We can realize a good prediction from the RF or ANN. However, we also need an explainable machine learning algorithm for the clinical application.
The random forest method is explained in the most accessible way possible, i.e., in the form of a colored decision tree, in which you can follow exactly how the program's decision-making process is going. Artificial neural networks are too complex for useful graphical interpretation.
- In the results section, the author mentioned 157 organ recipients. Among the total of 88 donors, 71 donors provided two kidneys, and 17 donors provided one kidney. However, there were 159 kidneys for the recipients (142 + 17 = 159). A discrepancy was noted. Please clarify.
We have corrected the mistake. Two donors from group were incorrectly assigned to the group of those who donated two kidneys. Ultimately, 69 donors donated 2 kidneys and 19 donated 1 kidney, for a total of 157 transplants.
- How many features should be imputed for the models? A continuous variable could be divided into category variables. For example, the BMI could be divided into underweight, normal, overweight, and obesity. Based on the eGFR level, the renal function status could be divided into normal kidney function and different stages of CKD (stage 3-5).
The number of variables for the model is limited by the computational complexity of the models and the practicality of the models in a clinical application. The huge number of variables makes it difficult to converge to the optimal solution. Too small number of variables may cause the descent to a solution that is not optimal too quickly. The original number of variables is recursively reduced towards the optimal subset. We wrote about the influence of the number of variables on the operation of machine learning algorithms in Konieczny et al. Our top 5 models used at least 4 variables common to all models and additional variables, as shown in Figure 2. All possible subsets of the input parameters we recursively selected, with a minimum size of 2 parameters. The best performance was achieved by models based on at least 4 key parameters: donor’s BMI, recipient’s BMI, recipient-donor weight difference and donor’s eGFR before procurement, Random Forest Classifier and MLP with Accuracy of 84.38%. Models with fewer input variables were completely ineffective. The best models we found required the above mentioned 4 variables plus EPTS, KDRI, KDPI, recipient's gender or recipient's age, and the result is the Random Forest and MLP models, which were summarized in Figure 2.
- Several important variables were not included in this study. For example, the time interval from donor to recipient and the time interval from donor’s brain death to the surgery could be considered. For the donor’s parameters, the comorbidity index or risk score for the severity of ICU hospitalization could be considered. For the recipient’s parameters, ABO mismatches, etiology of ESKD, dialysis parameter (kt/v, urea clearance rate, electrolyte, ca/p/PTH, baseline hemoglobin level, albumin, nutrition status), surgical time for kidney transplantation, and time interval from post-surgery to withdrawal from dialysis therapy.
We referred to the variables that were indicated as risk factors of DGF, in similar publications. The size of the original database, considering prognostic factors, was so large that it provided great opportunities for analyzing data, using machine learning, and selecting clinically significant ones. Nevertheless, the results found confirmation in the literature that the selected input parameters again turned out to be clinically significant. None of the models using the cold ischemia time variable achieved accuracy of more than 81.25%. The top model that used this variable at the input reached AUROC 0.78, Accuracy 81.25%, Precision 0.8318 (0.91 for non-DGF prediction and 0.60 for DGF presence), Recall 0.8125 (0.83 and 0.75 respectively) even though the literature shows a strong relationship between the occurrence of DGF and the prolonged CIT. We explained this discrepancy by the difference in methodology between classical odds ratios and random forest algorithms. Nevertheless, other key parameters are confirmed both in our work and in others.
Round 2
Reviewer 2 Report
No major comments after the improvements made by Authors
Reviewer 3 Report
All comments had been replied well and revised accordingly. I have no further suggestion.